# Prefix-Tuning Based Unsupervised Text Style Transfer

**Huiyu Mai**[1] [*], **Wenhao Jiang**[2] [†], **Zhihong Deng**[1]
[1]Peking University    [2]Guangming Laboratory
maihuiyu@pku.edu.cn, cswhjiang@gmail.com, zhdeng@pku.edu.cn

## Abstract

Unsupervised text style transfer aims at training a generative model that can alter the style of the input sentence while preserving its content without using any parallel data. In this paper, we employ powerful pre-trained large language models and present a new prefix-tuning-based method for unsupervised text style transfer. We construct three different kinds of prefixes, i.e., *shared prefix, style prefix*, and *content prefix*, to encode task-specific information, target style, and the content information of the input sentence, respectively. Compared to embeddings used by previous works, the proposed prefixes can provide richer information for the model. Furthermore, we adopt a recursive way of using language models in the process of style transfer. This strategy provides a more effective way for the interactions between the input sentence and GPT-2, helps the model construct more informative prefixes, and thus, helps improve the performance. Evaluations on the well-known datasets show that our method outperforms the state-of-the-art baselines. Results, analysis of ablation studies, and subjective evaluations from humans are also provided for a deeper understanding of the proposed method.

## 1 Introduction

Style transfer is an important task in computer vision and natural language processing. It aims to convert the input into the target style (e.g., season or artist style in image and sentiment or formality style in text) while preserving its content information. In addition, in text style transfer, the model also needs to ensure the fluency of the output sentence. Due to the lack of parallel data, the studies of text style transfer tasks mainly focus on unsupervised settings.

The mainstream unsupervised text style transfer (UTST) methods are based on the idea of disentangled representations of style and content. They usually use recurrent neural networks (RNNs) or Transformer (Vaswani et al., 2017) to encode style and content information of the input sentence with two independent embeddings. And the decoder takes the content embedding and the target style embedding as inputs to generate a new sentence with the expected content and style. But (Lample et al., 2018) has proved that it is not necessary to separate the style and content from the input sentence. The specially designed model can generate text with the target style by overwriting the original style. Therefore, we adopt a similar strategy in our model and directly feed the sentence and the target style information into the model for style transfer.

In previous works, the encoder encodes the sentence into a fixed-size, such as a 768-dimensional latent representation, which is small and hard to capture the full information of the sentence, leading to information loss. Thus, models without encoding the sentence to latent representation, e.g., decoder-only model, might be a better choice for the task of text style transfer. Recently, pre-trained auto-regressive language models have shown great success in text generation. Both GPT (Radford et al., 2019; Brown et al., 2020) and LLaMA (Touvron et al., 2023) use the decoder part in Transformer as their model framework. However, their ability in the task of UTST has not been fully investigated before. In this paper, we design a novel model to exploit auto-regressive model's high-quality text generation ability to achieve text transfer. Moreover, to avoid damaging the text generation ability, we adopt prefix-tuning paradigm (Li and Liang, 2021).

According to the above objectives, we propose a novel prefix-tuning framework based on GPT-2 for unsupervised text style transfer. Following previous works in the area of UTST (Zhu et al., 2017), we also employ the framework of adversarial learning due to the lack of parallel data. Thus,

---

[*] The work was done when the author was with Tencent as an intern.
[†] Corresponding author

our framework contains a generator and a discriminator. In the generator, three prefixes, i.e., *shared prefix, style prefix, and content prefix*, are proposed, and each of them serves different roles. The shared prefix provides task-specific information, which is independent of the input. The style and content prefix provide input-dependent information. The style prefix encodes the target style, and the content prefix extracts the content information in the input sentence. These prefixes provide information for GPT-2, guiding the model for style transfer. For the content prefix, no extra encoder is used. We simply use the pre-trained language model recursively to obtain the content prefix. In our model, the discriminator is also based on prefix-tuned language model. Therefore, only a fraction of the parameters of the whole model needs to be trained.

The contributions of our paper can be summarized as follows:

- We propose a new method based on prefix-tuning and pre-trained language model for unsupervised text style transfer. It is more expressive than previous methods, and only 2% of the parameters need to be trained.

- A strategy of recursively using language model to extract content information as content prefix that will be fed into the same LM for generation is proposed, which provide an effective way for the interactions between trainable and frozen parameters.

- Experiments conducted on well-known datasets show that our proposed method outperforms the state-of-the-art baselines. Evaluations by human and ablation studies are also provided to obtain a deeper understanding of our method.

## 2 Related Work

In this section, we present a brief summarization of topics related to our method.

### 2.1 Unsupervised Text Style Transfer

The existing UTST methods can be classified into three categories, i.e., prototype editing, pseudo-parallel corpus construction, and disentanglement method. Here we give a brief description of them.

A pipeline called Delete, Retrieve and Generate for prototype editing was proposed in (Li et al., 2018), which designed a process to revise the style markers in the target sentences. Works (Lee, 2020; Sudhakar et al., 2019) follow this pipeline, and they also employ transformer to improve performance.

Pseudo-parallel corpus construction methods use sentence embedding similarity to find sentence pairs with similar content but different styles from the corpus. Some works like LaMer (Liu et al., 2022) introduced scene alignment and reinforcement learning to improve performance. Other work (Luo et al., 2019) also used RL for UTST. IMaT (Jin et al., 2019) proposed the Iterative Matching and Translation method to construct a parallel corpus with high quality.

Disentanglement methods focus on separating the style and content of the input sentence. Cross-alignment (Shen et al., 2017) used a variational auto-encoder and a style classifier to encourage learning a style-independent content space. The reverse attention method was applied in the paper (Lee et al., 2021) to enhance content preservation. Some works also focus on adding the target style to the representation of the input sentence, like using a trained classifier to edit the latent representation obtained by the encoder through gradient-based optimization (Wang et al., 2019; Liu et al., 2020).

Since the ability of transformer has been demonstrated in many fields, some studies (Dai et al., 2019; Fan et al., 2022; Reif et al., 2022) used transformer-based encoder-decoder models. Style Transformer (Dai et al., 2019) is the most related work to ours. It treated the target style as an extra token, which is added to the input sentence, and the model modified the sentence according to the target style. However, it does not exploit the power of pre-trained models.

### 2.2 Prompt Learning Methods

Our method is also related to prompt learning, which is a class of methods that learn the input prompt with the pre-trained language models kept frozen. Prompt-learning based methods can exploit the knowledge obtained from the large-scale training data during the pre-training process. Since only the parameters of prompt are trainable, the number of parameters of prompt learning methods is usually quite small.

P-tuning was proposed in (Liu et al., 2021), and it enables GPT-2 to solve natural language understanding(NLU) tasks by employing trainable continuous prompt embeddings. It can automatically search prompts in the continuous space to bridge the gap between GPT-2 and NLU applications.

Rather than using embeddings, Prefix-tuning (Li

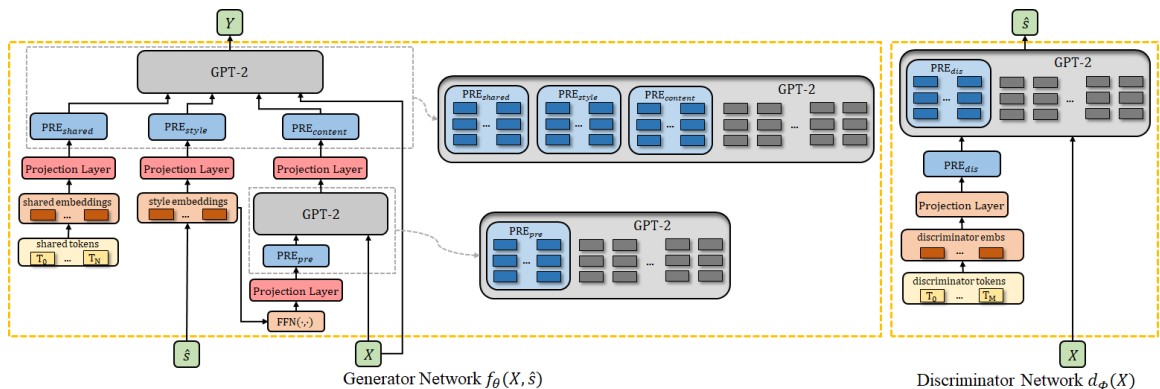

Figure 1: The architecture of the generator and discriminator. For the generator on the left, the *shared and style information* is first initialized as virtual token embeddings and then mapped to prefixes. The style embeddings are fed into $\mathsf{FFN}(\cdot,\cdot)$ together and passed into the projection layer to generate $\mathrm{PRE}_{pre}$. Then $\mathrm{PRE}_{pre}$ and the sentence $X$ are used as the input of GPT-2 to generate *content prefix*. Finally, the GPT-2 takes the three prefixes and $X$ as input for style transfer. The discriminator takes the prefix and $X$ as input and predicts its style $\hat{s}$. The orange blocks represent trainable embeddings and networks in our model. The pink "Projection Layer" blocks denote shared trainable parameters.

and Liang, 2021) uses key-value pairs in Transformer as prefixes, as a lightweight alternative to fine-tuning for the generation tasks, e.g., translation, summarization, and table-to-text. It uses a small number of trainable parameters to provide task-specific information for pre-trained language models, while still achieving comparable performance to fine-tuning the entire model.

Prompt-learning based methods have never been investigated for the setting of unsupervised text style transfer. In this paper, aiming at exploiting the knowledge in the pre-trained language models, we propose a new prompt-learning based method for unsupervised text style transfer.

## 3 The Proposed Method

In this section, we will describe the task settings of unsupervised text style transfer briefly and then present our method.

### 3.1 Problem Formalization

In this paper, we denote training set as $\mathcal{D} = \{(X_i, s_i)_{i=1}^{K}\}$, where $X_i = x_1, x_2, ..., x_n$ is a sentence, $n$ is the sentence length, and $s_i \in \{0, 1\}$ is the corresponding style label. The goal of text style transfer is to change the style of a sentence and keep its content. Specifically, the input sentence $X$ with style $s$ is converted into a new sentence $Y$ with a target style $\hat{s}$ with a model $f_\theta(X, \hat{s})$. In the unsupervised setting, there is no paired data $(X, Y)$ given, and the goal is to obtain the model $f$ with **only unpaired** training data $\mathcal{D}$.

### 3.2 Model Overview

We use GPT-2 (Radford et al., 2019) as our backbone. Rather than using the encoder-decoder model such as RNN or BART (Lewis et al., 2019) and T5 (Raffel et al., 2020), we choose a decoder-based model GPT-2, so as to encode the style and input text into prefixes. The overall framework of our model is shown in Fig. 1.

Like other UTST models, our model also adopts the framework of adversarial learning. As shown in Fig. 1, our model consists of a generator network and a discriminator network. The generator takes the input sentence $X$ and expected style $\hat{s}$ to generate the output sentence $Y$, and it is expected that the style of $Y$ is $\hat{s}$. The role of the discriminator is to predict the style of the input sentence.

In the generator, a special token, called style token $s$, is employed to provide the style information. Like other tokens, the style token $s$ is first transformed into an embedding, which is a single vector, with an embedding table. To obtain the style prefix for the language model, the style embedding is projected into $L$ vectors with a projection layer, which stands for the prefixes for the pre-trained language models with $L$ layers. To provide task-specific information for the pre-trained language model, we design $N$ shared token to achieve this goal. Similarly, the shared tokens are first transformed into $N$ embedding vectors, and then each of them is projected into prefixes for the $L$ layers. To reduce the number of parameters and facilitate the interaction among trainable parameters and frozen parameters, the style prefix and input sentence are fed into the same pre-trained language model to generate the

content prefix. In the discriminator, $M$ discriminator tokens are designed to provide information necessary for discrimination, and the procedure of prefix generation is similar to the shared prefix in the generator. The details of the proposed generator, the discriminator, and the training strategy will be described in the following subsections.

## 3.3 Prefix-Tuning Based Generator Network

The overall architecture of our generator $Y = f_\theta(X, \hat{s})$ is shown in Fig. 1. Three different prefixes, i.e., shared prefix, style prefix, and content prefix, are proposed to guide the style transfer of GPT-2. The prefix is the input, excluding the sentence tokens, for a certain layer of the pre-trained language model. Unlike the sentence tokens in the $l$-th layer are from the $l-1$-th layer, the proposed prefix for the $l$-th layer is from the projection layer. We will describe them in detail.

### 3.3.1 Shared Prefix

The role of the shared prefixes is to provide the pre-trained language model with task-related information. For specific text style transfer task son different datasets, the shared prefixes are expected to capture different information needed by the pre-trained language model.

In our model, the shared prefix is generated from $N$ shared tokens, denoted by $T_0, ..., T_N$. Similar to the ordinary tokens, the $N$ shared tokens are first transformed into $N$ shared embeddings, which are then projected into $N$ shared prefixes, denoted as $\mathrm{PRE}_{shared}$, with a feed-forward neural network. Each shared prefix is a set of $L$ vectors, where $L$ is the number layer of the pre-trained language model. The $l$-th prefixes vector is the input for the $l$-th layer. Note that the prefix for the $l$-th layer is not from the $l-1$-th layer of the language model. Instead, it comes from the projection layer directly.

### 3.3.2 Style Prefix

In previous works, the target style would be concatenated to the input sentence as a special token (Dai et al., 2019), or converted into latent representation and fed into the decoder (Lee et al., 2021). Unlike them, the target style is converted into a prefix and then fed into the generator in our method. Thus, the component for generating style prefixes contains more trainable parameters. Hence, it is more expressive. Therefore, using prefixes rather than embeddings can provide more information of style and is more beneficial to style transfer. The effects of the prefixes will be propagated upward

to all Transformer activation layers and rightward to input tokens. Similar to constructing the shared prefix, we use the projection layer to map the style embeddings to style prefix $\mathrm{PRE}_{style}$, where the style embeddings are derived from target styles.

### 3.3.3 Content Prefix and Recursively Applying LM

The role of the content prefix is to enhance the content information for the generation process. Even if the original sentence is also fed into the pre-trained language model, we found that adopting the proposed context prefix can improve the performance, which will be shown in the ablation study section.

The details about the generation of prefixes can also be found in Fig. 1. We can see that the same pre-trained model takes the $\mathrm{PRE}_{pre}$ and the input sentence $X$ as inputs, and the sentence tokens from the last layer are passed through a projection layer to produce the context prefix $\mathrm{PRE}_{content}$. The $\mathrm{PRE}_{pre}$ is a representation of style information. The embeddings of all styles are concatenated and are passed through a feed-forward neural network to reduce the dimensionality, followed by a projection layer to yield $\mathrm{PRE}_{pre}$. Finally, the three prefixes $\mathrm{PRE}_{shared}$, $\mathrm{PRE}_{style}$ and $\mathrm{PRE}_{content}$ are concatenated together as the prefix of the GPT-2 model to generate the sentence $Y$.

### 3.3.4 Generator Losses

Three different losses are adopted for the generator, including self-reconstruction loss, style transfer loss, and cycle generation loss. Fig. 2 shows the calculation of losses for our style transfer model. The training process of our model is similar to Cycle-GAN (Zhu et al., 2017) and Style Transformer (Dai et al., 2019), which are commonly used in unsupervised style transfer.

**Self reconstruction loss.** The generator takes the sentence $X$ and its original style $s$ as input so that we can train the generator to reconstruct the input sentence. This helps the model to maintain sentence content as much as possible, and the corresponding loss function can be expressed as:

$$\mathcal{L}_{self} = -\log P_\theta(X|X, s), \tag{1}$$

where $P_\theta()$ is the distribution over sequences from the generator.

**Style transfer loss.** The style quality of the generated sentence is supervised by the discriminator. Specifically, the sentence $X$ and the target style $\hat{s}$

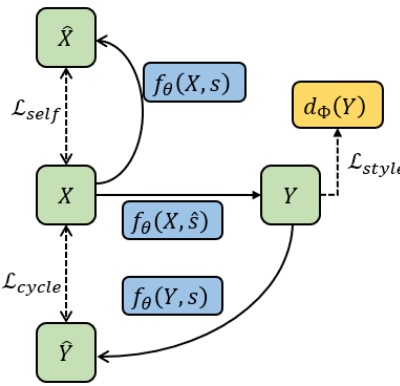

Figure 2: An illustration of losses adopted by the generator. The input sentence $X$ is transferred to $\hat{X}$ on the top and $Y$ on the right for calculating $\mathcal{L}_{self}$ and $\mathcal{L}_{style}$. $Y$ is passed into the generator again to generate $\hat{Y}$ for calculating $\mathcal{L}_{cycle}$.

are fed into the generator. Then the discriminator $d_\phi$ aims to maximize the probability that the generated sentence $Y$ belongs to the target style, and the corresponding loss can be expressed:

$$\mathcal{L}_{style} = -\log P_\phi(\hat{s}|Y). \tag{2}$$

**Cycle reconstruction loss.** Except for the above losses, we also adopt the cycle reconstruction loss to encourage the generator to preserve the content. Specifically, $Y$ and the original style $s$ are fed into the generator to generate $\hat{Y}$. Thus, $\hat{Y}$ is expected to be the same as $X$. So the loss is expressed as

$$\mathcal{L}_{cycle} = -\log P_\theta(X|Y, s). \tag{3}$$

The final loss is the weighted sum of the above three losses, which is

$$\mathcal{L}_{gen} = \lambda_1 \mathcal{L}_{self} + \lambda_2 \mathcal{L}_{cycle} + \lambda_3 \mathcal{L}_{style}, \tag{4}$$

while $\lambda_i$ is the weight of each loss. Both $\mathcal{L}_{self}$ and $\mathcal{L}_{cycle}$ help the style transfer model to achieve content preservation, and $\mathcal{L}_{self}$ also makes the model output as fluent as possible. And $\mathcal{L}_{style}$ helps models perform style transfer better.

Notice that the above three loss functions are only used to update the parameters of the generator, while the discriminator's parameters remain frozen. Since $\theta$ contains only the parameters used to construct the prefixes, we only need to train and save only 2% of the parameters of the pre-trained language model.

### 3.4 Discriminator Network

#### 3.4.1 Discriminator Structure

The discriminator aims to classify the sentence style from the generator, which will provide super-

vision for the generator. Specifically, if the dataset contains two styles, the discriminator is a classifier with three classes. The first two classes are the two styles, and the last class labeled as $s_{fake}$ represents that the sentence comes from the generator $f_\theta(\cdot, \cdot)$.

The architecture of $d_\phi(X)$ is shown on the right side of Fig. 1. To maintain the consistency of generation and discrimination, the discriminator $d_\phi(X)$ is also based on GPT-2 and prefix-tuning. It takes the prefix $\mathrm{PRE}_{dis}$ and $X$ as input, where $\mathrm{PRE}_{dis}$ is generated from token embeddings and projection layer just as the shared prefix. Then the average value of the last hidden layer is passed into a linear layer to calculate the probability that the sentence belongs to each category.

#### 3.4.2 Discriminator Loss

During training, the sentence $X$ comes from either the corpus or the generator. If $X$ comes from the corpus, the target $s_{target}$ is the style of the sentence. If $X$ is generated by the generator, the target is $s_{fake}$. The loss function is defined as follows:

$$\mathcal{L}_{dis} = -\log P_\phi(s_{target}|X), \tag{5}$$

where $P_\phi$ is the conditional distribution over styles obtained from the discriminator $d_\phi(X)$. Finally, the discriminator is updated while the generator remains frozen. Similar to generator training, the trainable parameter of the discriminator $\phi$ only includes the prefix part.

### 3.5 Training Strategy

During the learning process, the generator and discriminator are trained alternately. In our implementation, the discriminator is trained for ten steps, and the generator is updated for five steps in each iteration. Since prefix-tuning is employed, only the parameters for constructing prefixes will be updated, which account for a tiny part of the total model parameters.

## 4 Experimental Results

### 4.1 Datasets

We evaluate our method on two commonly used review datasets, i.e., the Yelp Reviews Dataset (Yelp)[1] and the IMDb Movie Review Dataset (IMDb)[2] (Maas et al., 2011). The Yelp dataset is a restaurant and business reviews dataset with

---

[1] https://www.yelp.com/dataset
[2] http://ai.stanford.edu/~amaas/data/sentiment/

| Dataset | Yelp | | IMDb | |
|---|---|---|---|---|
| | Positive | Negative | Positive | Negative |
| Train | 266,041 | 177,218 | 178,869 | 187,597 |
| Dev | 2,000 | 2,000 | 2,000 | 2,000 |
| Test | 500 | 500 | 2,000 | 2,000 |

Table 1: Statistic for Yelp and IMDb datasets.

sentiment labels provided by the Yelp Dataset Challenge. The IMDb dataset consists of movie reviews written by online users. Both datasets contain two styles, i.e., positive sentiment and negative sentiment. We followed the pre-processing procedure of Style Transformer (Dai et al., 2019) and used only highly polar sentences to train the model. The statistics of the two datasets after pre-processing are shown in Table 1.

## 4.2 Evaluation Metrics

Following the previous works (Dai et al., 2019; Fan et al., 2022), we evaluated our generated sentences from three perspectives, i.e., style, content, and fluency. In addition to the automatic evaluations, we also compare the performance of our method with other methods through human evaluation.

### 4.2.1 Automatic Evaluation

**Measure of style.** We measure the ability of style controlling of competing models by evaluating the accuracy of a commonly used classifier on the task of predicting the styles of generated sentences. Using fastText (Joulin et al., 2016), we train two classifiers for Yelp and IMDb on their training set, respectively, which are used the compute the style classification accuracy. A higher accuracy indicates that more output sentences have the same style as the target, which means that the model can transfer the sentence style better. The accuracy obtained is denoted as ACC in the rest of this paper.

**Measure of content.** We evaluate content preservation by calculating the BLEU score (Papineni et al., 2002) between the generated sentences and the original input sentences. A higher BLEU score means that the generated sentence retains more of the words in the original sentence, which indicates that the generating model achieves better content preservation. We also calculated the BLEU score between the generated sentence and the corresponding reference sentence for the Yelp dataset, which is provided by human. These two BLEU score metrics are referred to as self-BLEU and ref-BLEU in the rest of the paper, respectively.

**Measure of fluency.** The perplexity (PPL) of the generated sentence is adopted as the fluency metric. The model with lower PPL can generate more fluent sentences. We achieve this by training a 3-gram language model on the training set of two datasets using KenLM (Heafield, 2011).

### 4.2.2 Human Evaluation

Many previous works (Mir et al., 2019; Jin et al., 2022) have claimed that automatic evaluation methods have some shortcomings. Therefore, we also conduct a human evaluation to evaluate the results more accurately. For each dataset, we randomly select 200 sentences (100 for each class) from the test set. We will provide four annotators with the original sentences, the target style, and the output sentences of our model and other models. The annotators are asked to score the model output by considering the three aspects, i.e., style, content, and fluency. Scores for each aspect range from 1 (lowest) to 5 (highest). We use the average scores as the final scores.

## 4.3 Implementation Details

We use GPT2-large[3] as our base generator and discriminator, which contains 36 layers, 20 attention heads, and an embedding size of 1280. The shared and the style prefix lengths are 10 and 20, respectively. The size of our virtual token embedding is 1280, the same as the embedding size of GPT-2. The projection layer is a two-layer MLP with a hidden size of 128. We train our model on an NVIDIA Tesla V100 GPU with a batch size of 8. The Adam optimizer (Kingma and Ba, 2014) is used, and the learning rate is set to 0.0001. We tested different weights of the three losses on the validation set to find the best weight ratio. The experiments showed that our method is not sensitive to the three trade-off weights. The final weights of self-reconstruction loss, cycle loss, and classification loss are set to 0.25, 0.5, and 1.0, respectively.

## 4.4 Experimental Results
### 4.4.1 Automatic Evaluation

We compare our model with other disentanglement and prompt-based methods, including Deep Latent (He et al., 2020), Style Transformer (Dai et al., 2019), RACoLN (Lee et al., 2021), and LaMDA (Reif et al., 2022). The automatic evaluation result in Table 2 shows that our method significantly improved content retention and fluency.

---

[3] https://huggingface.co/gpt2-large

| Model | Yelp | | | | IMDb | | |
|---|---|---|---|---|---|---|---|
| | ACC(%)↑ | ref-BLEU↑ | self-BLEU↑ | PPL↓ | ACC(%)↑ | self-BLEU↑ | PPL↓ |
| Deep Latent | 83.3 | 18.0 | 50.9 | 153 | 77.9 | 60.8 | 147 |
| Style Transformer | 83.7 | 20.1 | 58.7 | 148 | 78.6 | 66.1 | 145 |
| RACoLN | 84.9 | 20.3 | 58.2 | 142 | 79.5 | 70.9 | 133 |
| LaMDA | **90.6** | 8.3 | 20.4 | **79** | / | / | / |
| Ours | 84.3 | **21.9** | **59.3** | 139 | **81.0** | **74.3** | **130** |

Table 2: Automatic evaluation results on Yelp and IMDb datasets. ACC is the accuracy of the style classifier on the generated sentences. ref-BLEU and self-BLEU are the content preservation metrics calculated by NLTK. PPL is the perplexity of the transferred sentences measured by KenLM.

| Model | Yelp | | | IMDb | | |
|---|---|---|---|---|---|---|
| | Style | Content | Fluency | Style | Content | Fluency |
| StyleTransformer | 3.9 | 4.2 | 4.3 | **3.6** | 4.2 | 4.1 |
| RACoLN | **4.0** | 4.5 | 4.2 | 3.5 | 4.3 | 4.1 |
| Ours | **4.0** | 4.6 | **4.5** | 3.5 | 4.6 | 4.2 |

Table 3: Human Evaluation results on both datasets. Each score is the average score from the four annotators. A higher score indicates better performance.

The style measuring metric ACC of our model on the Yelp dataset is competitive with other models, and on the IMDb dataset, it is higher than the baseline models. We find that our method outperforms other models on style control on the IMDb dataset, which has a long average sentence length. There are two main reasons. First, we use prefix-tuning to encode the target style, which contains richer information than the latent representations used by other methods, and thus can better cope with longer texts. In addition, our discriminator also uses GPT-2, which can handle longer texts better than RNN, thus achieving better style control.

Our model shows a noticeable improvement in the BLEU score, especially the ref-BLEU score on the Yelp dataset, which is 1.6 higher than RACoLN, a relative improvement of 7%. And our self-BLEU score on IMDb is 3.4 higher than the baselines. We attribute this improvement to content prefixes. On the one hand, the content prefix contains richer information than the latent representation. On the other hand, by recursively using GPT-2, there is more sufficient interaction between the input text and the language model so that it can obtain more content information. At the same time, since GPT2-large is used, the outputs are more fluent and have lower perplexity.

LaMDA (Reif et al., 2022) is a zero-shot method based on large language model (LLM). As can be seen in the table, it achieves considerable ACC results. In addition, using large language model

helps it output more fluent sentences, improving the PPL metric. However, since these methods do not introduce any supervision, they perform poorly in content preservation.

### 4.4.2 Human Evaluation

The human evaluation results are shown in Table 3. We choose two other methods to compare with ours. As seen from the table, the human evaluation results are basically consistent with the automatic evaluation results. Our model outperforms the other two models in content preservation and fluency while keeping style control metrics close to those of the other two models. It can be seen from the results of the automatic evaluation and human evaluation that our method can better preserve the content of the original text and ensure the fluency of the output sentence while also achieving a good style transfer effect.

### 4.5 Ablation Study

Ablation studies are performed to investigate the impacts of each module on performance. We remove each prefix by turns and retrain our model while keeping other settings unchanged. To compare the performance of using different ways to encode styles, a model using style embeddings instead of prefixes is also designed. Lastly, a model with all parameters fine-tuned was trained for comparison with prefix-tuning. These models are evaluated on Yelp, and the results are shown in Table 4.

Without the shared prefix, our model shows a slight drop in style control and content preservation metrics. This indicates that it loses part of the task information because the shared prefix is a prompt of the whole task. In addition, the shared prefix also provides common information between different styles, which is also beneficial to style transfer.

Removing the style prefix has a significant impact on style transfer. Without the style information, the model easily degenerates into a model that

| Model | ACC↑ | r-BLEU↑ | s-BLEU↑ | PPL↓ |
|---|---|---|---|---|
| - Shared Prefix | 82.1 | 21.2 | 55.7 | 152 |
| - Style Prefix | 25.8 | 23.3 | 82.5 | 130 |
| + Style Embedding | 82.2 | 21.3 | 59.4 | 135 |
| - Content Prefix | 84.0 | 18.5 | 44.1 | 167 |
| Ours | 84.3 | 21.9 | 59.3 | 139 |
| Ours + Full fine-tune | 85.0 | 22.3 | 59.6 | 139 |

Table 4: Ablation studies on the Yelp dataset. (-) indicates that this part of the prefix is removed from our model. r-BLEU and s-BLEU indicate ref-BLEU and self-BLEU, respectively.

directly copies the input sentence without modifying the style. Therefore, the model performs better on the BLEU score than the full model, with a significant drop in the ACC score. The performance of the model using style embedding on ACC metric is worse than using the prefix, indicating that style prefix can provide more style information and improve the performance of the model.

After removing the content prefix, the self-BLEU score significantly drops by 15.1. There are two main reasons that the content prefix can improve performance. First, it provides more information about the input sentence for the generator, which helps the model retain content information better. Secondly, the input sentence will be passed into GPT-2 twice during style transfer, which allows GPT-2 to fully extract content information and achieve better content preservation.

Fine-tuning the whole model can be regarded as the upper bound of the performance. Experimental results show that using prefix-tuning can achieve close results to training the whole model. Since we only need to train 2% parameters compared to GPT-2, this performance is quite competitive.

### 4.6 Case Study

Finally, to better understand the characteristics of different models, several sentences are sampled from the Yelp dataset and tested on different models, including Style Transformer, RACoLN, and ours. The outputs are shown in Table 5. Some sentences generated by the baselines appeared incoherent or made some grammar errors, which do not appear in ours. This is mainly due to the backbone we use. The GPT-2 with solid expressive ability makes the output sentences of outputs more fluent. In addition, these grammar errors in the baseline methods also affect the content preservation of the original input, which results in lower BLEU scores. On style transfer, the baseline models miss some words that need to be modified, or

| **Negative to Positive** | |
|---|---|
| Input | There chips are ok, but their salsa is really bland. |
| StyleTrans | There chips are ok, but their salsa is really outstanding. |
| RACoLN | There chips is delicious, and their salsa is really really. |
| Ours | There chips are ok, and their salsa is really delicious. |
| Input | The wine was very average and the food was even less. |
| StyleTrans | The wine was very average and the food was even less delicious wine. |
| RACoLN | The wine was very average and the food is even good. |
| Ours | The wine was very excellent and the food was even better. |
| **Positive to Negative** | |
| Input | I love this place, the service is always great! |
| StyleTrans | I guess this place, the service is not great! |
| RACoLN | I avoid this place, the service is always horrible! |
| Ours | I hate this place, the service is always bad! |
| Input | The best mexican food in the phoenix area. |
| StyleTrans | The this mexican food in the phoenix area. |
| RACoLN | The great mexican food in the phoenix area. |
| Ours | The worst mexican food in the phoenix area. |

Table 5: Some examples generated by the models on Yelp dataset. The red words indicate the grammar error. The blue words indicate bad transfer.

replace the words with the wrong words, such as "but" in the first sentence and "best" in the last. The possible reason is that both of them use embeddings to represent the target style. Different from them, the target style is encoded as the style prefix in our model, which contains more parameters and can significantly improve style transfer quality.

## 5 Conclusion

In this paper, a novel method for unsupervised text style transfer based on prefix-tuning was proposed, in which we take advantage of the pre-trained language model. We encode both target style and input text with prefixes that guide style transfer using prefix-tuning. We design a style transfer method that uses the language model recursively, using GPT-2 in both the content extraction and style transfer stages. In this way, the input sentence can fully interact with the language model, fully utilizing the language model's capabilities for style transfer. Experiments show that our method significantly improves content preservation and fluency of the style-transferred sentences compared to the baselines while achieving competitive results on the style control. In the future, we plan to extend our approach to more broad application settings.

## Limitations

The proposed method needs to be trained for new style transfer tasks. Thus, we will explore combining the concept of in-context learning in the future and employ the more powerful pre-train models, i.e., ChatGPT, to improve the performance and extend it to the zero-shot setting.

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
