# OpenReview forum: "Prefix-Tuning Based Unsupervised Text Style Transfer"
_EMNLP/2023/Conference — EMNLP 2023 Findings_

### Official Review · Reviewer_cYMW · 2023-08-03

**Soundness:** 3

**Excitement:**

2: Mediocre: This paper makes marginal contributions (vs non-contemporaneous work), so I would rather not see it in the conference.

**Missing References:**

- Style Transfer as Unsupervised Machine Translation. Zhang et al., 2018.
- A Dual Reinforcement Learning Framework for Unsupervised Text Style Transfer. Luo et al., 2019.

**Paper Topic And Main Contributions:**

This paper focuses on unsupervised text style transfer, which involves modifying the style of input sentences while preserving their content, without relying on parallel data. Building on the framework of (Dai et al., 2019), the authors aim to utilize pre-trained language models (such as GPT-2) and prompt learning techniques to construct generators and discriminators. Furthermore, the paper adopts a recursive approach to improve the interactions between the input sentence and GPT-2. The proposed method is evaluated on Yelp and IMDb datasets, and the experimental results demonstrate its effectiveness.

**Questions For The Authors:**

- Can the authors provide more details on the specific differences between their proposed method and the framework presented in (Dai et al., 2019)? How does the incorporation of prompt learning techniques contribute to the proposed method?
- While the evaluation of the proposed method focuses on sentiment transfer tasks, are there any plans to explore its performance on other tasks, such as formality transfer? If so, what are the potential challenges and considerations in adapting the method to different tasks?
- The experiments in this paper only compare a limited number of baselines. Are there any specific reasons for choosing these particular baselines? Would it be possible to include additional baselines, such as DelRetri [Li et al., 2018], Template [Li et al., 2018], UnsuperMT [Zhang et al., 2018], and DualRL [Luo et al., 2019], to provide a more comprehensive comparison?

**Reasons To Accept:**

- Proposing a new method based on GPT-2 for unsupervised text style transfer.
- Introducing three different types of prefixes to encode task-specific information, target style, and content details, thereby providing more comprehensive information to the model compared to the embeddings used in previous works.
- Adopting a recursive approach to improve the interactions between the input sentence and GPT-2.
- Providing comprehensive results, conducting ablation studies, and including subjective evaluations from human assessors to gain a deeper understanding of the proposed approach.

**Reasons To Reject:**

- This paper presents incremental work. The proposed method is highly similar to the framework of (Dai et al., 2019) and incorporates prompt learning techniques that have been previously proposed.
- The evaluation of the proposed method is limited to sentiment transfer tasks. It would be beneficial to explore the performance of the proposed method on other tasks (such as formality transfer) and datasets.
- The experiments in this paper only compare a small number of baselines. It would be advantageous to include additional baselines, such as DelRetri [Li et al., 2018], Template [Li et al., 2018], UnsuperMT [Zhang et al., 2018], and DualRL [Luo et al., 2019].

**Reproducibility:**

4: Could mostly reproduce the results, but there may be some variation because of sample variance or minor variations in their interpretation of the protocol or method.

**Reviewer Confidence:**

5: Positive that my evaluation is correct. I read the paper very carefully and I am very familiar with related work.

---

> ### Author Rebuttal · Authors · 2023-08-29
>
> Thanks for the valuable comments. Here are our responses.
>
> **1. Differences with Style Transformer (Dai et al., 2019).**
>
> Our method is based on the core idea to employing the knowledge from the pre-trained and frozen model with prefix-tunning. Thus, the design of prefixes is important to achieve such a purpose. We proposed 3 prefixes to encode content, style and task-specific information. Moreover, we also proposed to use LM recursively to provide an effective way for the trainable and frozen parameters to interact. Thus, our method is totally different from Style Transformer.
>
> **2. Evaluation on more complex tasks.**
>
> We agree with the reviewer that it would be interesting to evaluate our method under more complicated settings. It seems that our method is not restricted to the task of sentiment transfer. In this paper, we aim to illurstrating the method clearly, which is the reason why we used relatively simple tasks to evaluate our method. We are still working on evaluating it on more complex tasks, and the results will be included in the extension version of our paper.
>
>
> **3. Compare with more methods.**
>
> Thanks for the suggestion. The methods in [Li et al., 2018] have been compared in Style Transformer, and our method is better than the Style Transformer. Thus, our method should be better than the methods in (Li et al., 2018). We will include the discussions and analysis of all the methods mentioned here in our revised version.

---

### Official Review · Reviewer_p4xo · 2023-08-04

**Soundness:** 4

**Excitement:**

4: Strong: This paper deepens the understanding of some phenomenon or lowers the barriers to an existing research direction.

**Paper Topic And Main Contributions:**

This paper proposes an unsupervised text style transfer method which is an adversarial framework consisting of a generator and a discriminator. Both the generator and discriminator are using GPT-2 as base models. Moreover, the authors use prefix-tuning to finetune the models. They design shared prefix, style prefix, and content prefix to encode corresponding information of the input sequence. The proposed method outperforms baselines in terms of both automatic evaluation and human evaluation from the aspects of style, content, and fluency.

**Reasons To Accept:**

1.	This work is clearly motivated to do text style transfer and proposes a quite new and interesting.
2.	The adversarial framework of the methodology is reasonable. The prefixes are quite significant according to the experimental results. Moreover, the losses designed for the generator are quite thoroughly considered.
3.	The statement of the methodology and the structure of the paper are good. The implementation is comprehensive. The results are solid.


**Reasons To Reject:**

1.	Baselines in this paper are limited. There are no baselines from 2022. There a also no baselines of editing-based [1] or prompt-based methods [2].
2.	There is no analysis of the three losses to train the generator.

[1] Machel Reid and Victor Zhong. 2021. LEWIS: Leven- shtein editing for unsupervised text style transfer. In Findings of ACL-IJCNLP, pages 3932–3944.
[2] Mirac Suzgun, Luke Melas-Kyriazi, and Dan Jurafsky. 2022. Prompt-and-Rerank: A Method for Zero-Shot and Few-Shot Arbitrary Textual Style Transfer with Small Language Models. In Proceedings of the 2022 Conference on Empirical Methods in Natural Language Processing, pages 2195–2222.


**Reproducibility:**

3: Could reproduce the results with some difficulty. The settings of parameters are underspecified or subjectively determined; the training/evaluation data are not widely available.

**Reviewer Confidence:**

3: Pretty sure, but there's a chance I missed something. Although I have a good feel for this area in general, I did not carefully check the paper's details, e.g., the math, experimental design, or novelty.

---

> ### Author Rebuttal · Authors · 2023-08-29
>
> Thanks for the valuable comments. Here are our responses.
>
> **1. More baselines should be compared.**
>
> Thanks for pointing out that. We will include the comparison with related methods under the same settings as ours in the revised version.
>
> **2. There is no analysis of the three losses to train the generator.**
>
> Thanks for the suggestion. The trade-off parameters in Eq.(4) (0.25, 0.5 and 1) are selected on the valid dataset. We just fix the first one and tune the others by multiplying a factor of 2. And they are not carefully tuned. Our experiments showed that our method is not sensitive to the three trade-off parameters. We will add the descriptions in the revised version of our paper.

---

### Official Review · Reviewer_2nkN · 2023-08-13

**Soundness:** 3

**Excitement:**

3: Ambivalent: It has merits (e.g., it reports state-of-the-art results, the idea is nice), but there are key weaknesses (e.g., it describes incremental work), and it can significantly benefit from another round of revision. However, I won't object to accepting it if my co-reviewers champion it.

**Paper Topic And Main Contributions:**

The paper suggested a method of unsupervised text style transfer with prefix tunning approach. they followed adversarial learning approach where they have a generator and discriminator. For generator, they trained three types of prefixes (style, content, and task) and for the discriminator they also trained prefix tokens. Prefix tuning is beneficial to reduce the number of trainable parameters in both generator and discriminator.
The paper attempts to transfer styles between texts but they didn't describe the problem clearly - hence readers are assumed to understand why the problem is needed from their past experience (e.g., why do we need text style transfer from the first place? how encoder leads to information loss? why generative model is important? .... etc. ).

**Reasons To Accept:**

* The method provided improvements compared to three different models. Compared to LaMDA, the improvement is mainly in BLEU score.
* The paper is written clearly (but there are parts that need to be explained better (e.g., why do we need the recursive exactly and what happens if we  do not do that?)

**Reasons To Reject:**

* The paper is only evaluated on two datasets that target the same task (positive to negative sentences or vice versa). The approach would be more powerful if different types of styles have been considered. The provided examples focus on changes for the language (e.g. but) or antonym changes (e.g. love vs. hate) but not clear how does it perform in more complex scenarios.
* The paper has focused on transferring only two styles to each other. Not clear what happens with scaling the number of styles.

**Reproducibility:**

3: Could reproduce the results with some difficulty. The settings of parameters are underspecified or subjectively determined; the training/evaluation data are not widely available.

**Reviewer Confidence:**

3: Pretty sure, but there's a chance I missed something. Although I have a good feel for this area in general, I did not carefully check the paper's details, e.g., the math, experimental design, or novelty.

---

> ### Author Rebuttal · Authors · 2023-08-29
>
> Thanks for the valuable comments. Here are our responses.
>
> **1. Evaluated only on two datasets and two-style transfering task, which are relatively simple.**
>
> We agree with the reviewer that it would be interesting to evaluate our method under more complex settings. Actually our method is not restricted to the task of sentiment transfer. In this paper, we aim to illurstrate the method clearly. Thus, we used relatively simple tasks to evalute our method. We are also working on evaluating it on more complex tasks, and the results will be included in the extention verion of our paper.
>
> **2. Did not describe the problem clearly.**
>
> We will describe the necessary background, motivations for each step of our method more clearly in the revised version.

---

### Meta-Review · Area_Chair_d4Q5 · 2023-09-17

**Recommendation:** 3

**Metareview:**

The paper presents a method for unsupervised style transfer, using prefix-tuning of GPT-2 with 2 types of prefixes: style, content, and task. The proposed approach was properly evaluated using both automatic metrics and human evaluation, as well as ablation studies, but it was not evaluated against recent and similar models. Another concern is that the work seems incremental and the results are only slightly better than the baselines, and the illustrated examples are probably cherry-picked to favor the proposed model. In addition, it was only evaluated on sentiment task (for transferring sentiment from positive no negative and vice versa), so it's not clear how well it would work for other types of style transfer. Overall, while the paper is well written and the approach is somewhat sound, the paper needs extensive evaluations to validate the method for different style transfer tasks compared to other recently proposed approaches.

---

### Decision · Program_Chairs · 2023-10-07

**Decision:**

Accept-Findings

**Comment:**

The paper presents a method for unsupervised style transfer, using prefix-tuning of GPT-2 with 2 types of prefixes: style, content, and task. The proposed approach was properly evaluated using both automatic metrics and human evaluation, as well as ablation studies, but it was not evaluated against recent and similar models. Another concern is that the work seems incremental and the results are only slightly better than the baselines, and the illustrated examples are probably cherry-picked to favor the proposed model. In addition, it was only evaluated on sentiment task (for transferring sentiment from positive no negative and vice versa), so it's not clear how well it would work for other types of style transfer. Overall, while the paper is well written and the approach is somewhat sound, the paper needs extensive evaluations to validate the method for different style transfer tasks compared to other recently proposed approaches.